

# Technical note: A microcontroller-based automatic rain sampler for stable isotope studies

Nils Michelsen[1], Gerrit Laube[2], Jan Friesen[3], Stephan M. Weise[3], Ali Bakhit Ali Bait Said[4], Thomas Müller[2]

[1]Institute of Applied Geosciences, Technische Universität Darmstadt, Darmstadt, 64287, Germany
[2]Department of Hydrogeology, UFZ Helmholtz Centre for Environment Research, Leipzig, 04318, Germany
[3]Department of Catchment Hydrology, UFZ Helmholtz Centre for Environment Research, Halle, 06120, Germany
[4]Ministry of Regional Municipalities and Water Resources, Salalah, Sultanate of Oman

*Correspondence to*: Nils Michelsen (michelsen@geo.tu-darmstadt.de)

**Abstract.** Automatic samplers represent a convenient way to gather rain samples for isotope ($\delta^{18}$O and $\delta^2$H) and water quality analyses. Yet, most commercial collectors are expensive and do not reduce post-sampling evaporation and the associated isotope fractionation sufficiently. Thus, we have developed a microcontroller-based automatic rain sampler for timer-actuated collection of integral rain samples. Sampling periods are freely selectable (minutes to weeks) and the device

is low-cost, simple, robust, and customizable. Moreover, a combination of design features reliably minimizes evaporation from the collection bottles. Evaporation losses were assessed by placing the pre-filled sampler in a laboratory oven with which a diurnal temperature regime (21-31°C) was simulated for 26 weeks. At the end of the test, all bottles had lost less than 1 % of the original water amount and all isotope shifts were within the analytical precision.

These results show that even multi-week field deployments of the device would result in rather small evaporative mass

losses and isotope shifts. Hence, we deem our sampler a useful addition to devices that are currently commercially available and/or described in the scientific literature. To enable reproduction, all relevant details on hard- and software are openly accessible.

## 1 Introduction

The stable isotopes $^{18}$O and $^2$H represent nearly ideal tracers that are frequently applied in hydrology and other disciplines.

Yet, most applications require data on the isotope composition of precipitation (Bowen and Revenaugh, 2003; Hughes and Crawford, 2013).

Although isotope analyzers have become field-deployable, enabling high-resolution on-site rain analyses (Berman et al., 2009; Herbstritt et al, 2018a; Munksgaard et al., 2011, 2012; Pangle et al., 2013; von Freyberg et al., 2017; Windhorst et al., 2017), many researchers hesitate to take their analyzer to the field due to the risk of damaging the expensive device and due

to logistical constraints such as high power demand (Herbstritt et al., 2018b). Hence, most samples are still obtained in more traditional ways for subsequent analysis in the laboratory. They are collected manually (e.g., in intra-event studies; Conroy et al., 2016; Michelsen et al., 2015), with cumulative rain collectors (summarized in Michelsen et al., 2018), or with automatic samplers.

Such automatic samplers, initially mostly developed for precipitation chemistry studies, cover a broad range of designs (see

reviews by Krupa, 2002; Laquer, 1990; Robertson et al., 1980). Simple versions of the "linked collection vessels" type (Robertson et al., 1980), featuring a set of serially connected bottles that are consecutively filled, are inexpensive and easy to construct. They have no moving parts and collect samples on a volume basis (Laquer, 1990). Particularly the designs by Kennedy et al. (1979) and Vermette and Drake (1987) are still used nowadays (Fischer et al., 2016, 2017; Hervé-Fernández et al., 2016; Muñoz-Villers and McDonnell, 2013; Saffarpour et al., 2016), albeit sometimes with modifications (Buda and

DeWalle, 2009; Qu et al., 2017). Drawbacks of this sampler type comprise the lack of an efficient evaporation reduction



mechanism, possibly causing post-sampling fractionation (Fischer et al., 2017) and a potential for cross-contamination of samples.

More complex "automatically segmenting samplers" (Robertson et al., 1980) are commercially available, but usually expensive (several thousand €; Tauro et al., 2018). Further, they often do not minimize evaporation sufficiently (Hartmann et al., 2018; Tauro et al., 2018; Williams et al., 2018). To overcome the latter problem, researchers have occasionally added paraffin oil to the collection bottles of such commercial samplers (Birkel et al., 2011; van Huijgevoort et al., 2016; Sprenger et al., 2017; Tunaley et al., 2017), as recommended by Fergusson (1921). However, a quantitative oil removal is not easy and oil traces might cause problems during laboratory analyses (Gröning et al., 2012), particularly in laser spectroscopy (IAEA, 2014). Fischer et al. (2016) thus explicitly avoided the oil method and preferred to empty their automatic samplers directly after rainfall events.

The isotope community also developed various custom-made automatic samplers, mostly for high-resolution sampling of rain and/or surface water (e.g., Camacho Suarez et al., 2015; Conroy et al., 2016; Coplen et al., 2008; Muller et al., 2015; Neuhaus, 2016, 2018; Siebert, 2015; Terzer-Wassmuth et al., 2018). While some of these collectors are outlined in the corresponding patent specifications (Coplen, 2010; Neuhaus, 2016; Siebert, 2015), several others are only briefly described or not at all. We can imagine that, at least in some of these cases, the researchers did not provide details, because they saw their collectors as means to an end, i.e., as tools helping to generate data on which they eventually concentrated.

Notable exceptions in this context are Hartmann et al. (2018) and Nelke et al. (2018), who do provide sufficient details on their devices to enable reproduction. The former authors use a peristaltic pump to inject water directly into air-tight vials. Nelke et al., by contrast, utilize two peristaltic pumps to direct water into aluminium-lined bags using solenoid valves. Both collectors apparently focus on continuously flowing media (e.g., dripwater, surface water; instead of discontinuous media such as precipitation) and take discrete "snapshot" samples (instead of integral samples) of small to moderate size (12 and 250 mL, respectively). Moreover, their designs have in common that they are rather sophisticated. Yet, complexity can be a double-edged sword. While their technical solutions are certainly elegant, the advanced designs might be somewhat difficult to reproduce.

Here, we describe a complementary automatic rain sampler. Our simple, robust, and low-cost collector allows the timer-actuated collection of integral rain samples, with time intervals ranging from minutes to weeks. A combination of design features effectively reduces post-sampling evaporation. In the spirit of Open Science (cf. Hartmann et al., 2018; Nelke et al., 2018), we provide a detailed description of our customizable collector and its components. Moreover, we present the results of a 26-week test addressing the evaporation reduction capacity of the device in a warm climate.

## 2 Design

The following section gives an overview of the sampler design. Further details (bill of materials, technical drawings, circuit diagram, code, and manuals) enabling reproduction are provided on the website https://www.ufz.de/index.php?en=44048 (section Documentation).

The sampler collects rain by a funnel from which the water flows into a distribution unit, the core of the device (Fig. 1). It consists of two custom-made uniaxial discs (separated by a 2 mm neoprene rubber seal) with drill holes that are positioned opposite of each other. The upper disc (PTFE) is the rotor (rotates clockwise) and has two drill holes fitted with push-in ports. The outer port is the rain inlet and the inner port is the air outlet. The lower disc (PVC-U) is the stator (remains static) and features 36 ports that are arranged in two circles. The ports of the outer circle are connected to water tubes and the ports of the inner circle are connected to air tubes (LDPE; see section Pre-test of tubing materials in the Supplement, incl. Fig. S1, Table S1).





Water coming from the funnel flows through the rain inlet into the distribution unit and then through a water tube, into the first 500 mL sampling bottle (HDPE; cf. Spangenberg et al. 2012). To minimize post-sampling evaporation, we adapted the concept of an evaporation-free cumulative collector (Gröning et al., 2012), which has been successfully tested under hot and arid conditions (Michelsen et al., 2018) and is advocated by the IAEA (2014). In this concept, an inlet tube reaches the

bottom of the bottle. A few millimeters of rainfall are sufficient to cause a water level rise into the tube, thus decoupling the bottle headspace from the atmosphere. This process can be amplified by inserting the end of the inlet tube into a small container (cf. Gröning et al., 2012; or a short piece of bent tubing, see Fig. 1), reducing the amount of rain needed to decouple the air in the bottle from the atmosphere. The air displaced by the inflowing water leaves the bottle through the air tubing. The latter leads from the bottle cap back to the distribution unit. Here, the air is pushed through the air outlet into a

15 m long pressure equilibration tube (cf. Gröning et al., 2012, IAEA, 2014).

At the end of the freely selectable sampling interval, a microcontroller triggers a stepper motor to turn the rotor of the distribution unit by 20°. Through this rotation, the first sampling bottle is isolated from the atmosphere – the rotor disc closes the water and air tubes of this bottle. In turn, rain can now flow into the second bottle. The air thus displaced from the second bottle is pushed into the aforementioned pressure equilibration tube, i.e., this tube is shared by all bottles. A tight fit of the

rotor is ensured through a spring above the disc, providing sufficient pressure to avoid leakages.

The microcontroller features a display and buttons for convenient programming, avoiding the necessity of a notebook during field setup or maintenance. Moreover, the microcontroller has a set of low power modes that are excessively used. In fact, the microcontroller and the accompanying stepper motor driver are only active during initial setup and between sampling periods, while the distribution unit is driven to the next sampling port. The fact that the distribution unit consumes virtually

no power during sampling, allows for week-long sampling periods on just two AA batteries for the logic unit and another eight AA batteries for the stepper motor.

The stepper motor and the control unit (microcontroller with accessories and batteries) are each located in a dust- and water-proof enclosure and connected by a water-proof cable (IP68). This robust design loosens the requirements for the overall enclosure. For field deployments (not presented here), a simple plastic storage box was chosen to be sufficient to house the

sampling bottles and the automatic collector.

## 3 Evaporation experiment

### 3.1 Methods

After initial tests targeting the functional capabilities of the sampler (timing, rotation angles, etc.), we studied the evaporation reduction efficiency of the device.

To this end, five HDPE bottles (500 mL; Labsolute, Renningen, Germany) were partially filled with water of known isotopic composition ($\delta^{18}O$=-8.53 ‰, $\delta^{2}H$=-60.7 ‰ related to Vienna Standard Mean Ocean Water, V-SMOW) and connected with water and gas tubing to the distribution unit (Table 1). The latter was not coupled to the microcontroller unit, i.e., the rotor did not move during the test.

Bottle 1 (filled with 100 mL) was connected to the open ports, i.e., its water and air tubes led to the ports beneath the rain

inlet and the air outlet, respectively (Fig. 2). A 2 m long rain tube (without funnel) was connected to the rain inlet and a 15 m long pressure equilibration tube was connected to the air outlet. With this bottle, we wanted to assess how long a sampling bottle can be left "exposed" to the atmosphere (exposed via the rain tube and the pressure equilibration tube).

Bottles 2 to 5 contained different water amounts (100, 200, 300, and 400 mL) and their water and air tubes led to stator ports that were blocked by the rotor disc. Here, the goal was to determine how long a sample can remain in the collector without

undergoing critical evaporative mass loss.





This setup was placed for 26 weeks (cf. Hartmann et al., 2018) in a laboratory drying oven (T 6120 by Heraeus, Hanau, Germany) with which a diurnal temperature regime was simulated (Michelsen et al., 2018). A socket timer triggered a daily 12 h heating period (31°C). After this phase, the oven was allowed to cool down to room temperature (approx. 21°C). To accelerate this cooling process, the oven was opened daily for three hours. Temperature and relative humidity in the oven

were logged in 10 min intervals (DK320 HumiLog ruggedPlus by Driesen + Kern GmbH, Bad Bramstedt, Germany).

For comparison, two additional bottles were placed into the oven – Bottles 6 and 7 (both filled with 100 mL). Bottle 6 was closed, thus representing a best case scenario. With this approach, potential diffusion through the bottle material was tested. Bottle 7, by contrast, was "open", i.e., it featured two holes (diameter 6 mm) in its cap, to represent the worst case (no evaporation suppression).

Moreover, three identical HDPE tubing loops (häberle LABORTECHNIK, Lonsee-Ettlenschieß, Germany) were included in the oven experiment. They were partially filled with 0.5 mL of water and the tubing ends were connected to each other with a connector. All three loops had a circumference of 25 cm, which corresponds to the lengths of the tubing between the bottles and the distribution unit. This allowed for an estimate of diffusive water fluxes through the tubing material.

To determine evaporative mass losses during the experiment, the bottles and the tubing loops were weighed repeatedly

(Voyager Pro VP2102C by Ohaus, Pine Brook, USA). Bottles 1 through 5 were disconnected from the distribution unit for this purpose. Samples for isotope analyses (1.6 mL in 2 mL vials) were gathered after 6, 16, and 26 weeks. In case of Bottles 1 to 5, water was withdrawn through the water tube with a syringe. Bottle 6 was sampled through a gas-tight sampling port in its cap (Mininert by VICI Precision Sampling, Baton Rouge, LA, USA) with a syringe. Samples from Bottle 7 were taken with a Pasteur pipette through one of the holes in the cap. To account for these artificial mass losses, the bottles were

weighed before and after sampling.

At the end of the test, the obtained water samples were analyzed for their isotopic composition by Laser Cavity Ring-Down Spectroscopy (L2130-i by Picarro, Santa Clara, CA, USA). The results were expressed in per mil (‰) using the conventional delta-notation relative to V-SMOW. The external precisions (±1σ), determined by repeated analyses of a control sample, were ±0.15 ‰ and ±0.6 ‰ (±1σ), for $\delta^{18}O$ and $\delta^2H$, respectively.

### 3.2 Results

#### 3.2.1 Evaporative mass losses

The applied diurnal temperature regime (21-31°C; see Fig. S2) mostly resulted in small but measurable evaporative mass losses that increased over time (Table 1).

After 26 weeks, absolute mass losses ranged between 0.8 and 2.4 g for the bottles connected to the distribution unit (Bottles

1 through 5). Bottle 6 (closed), representing the best case, showed a lower mass loss of 0.24 g and in case of Bottle 7 (open; worst case), a loss of 24.42 g was encountered. The three tubing loops (each 25 cm long) lost about 0.14 g over the same time period (see Table S2).

These data suggest that diffusive losses through the tubing material of the connected bottles (two tubes per bottle, hence 0.28 g) and through the bottle material (0.24 g) are similar. The connected bottles appear to have additional leakages, but the

overall absolute losses are still rather small, particularly when compared to the worst case scenario, an unprotected bottle (Bottle 7).

This becomes clearer when the data are put in perspective with the original water amounts. Fractional losses of Bottles 1 through 5 range from 0.40 to 0.94 %-orig, i.e., even after half a year the maximum loss was below 1 %. As expected, these values are somewhat higher than in case of the closed bottle (Bottle 6; 0.24 %-orig), but far below the loss recorded for the

open Bottle 7, which lost nearly a quarter of its water (24.42 %-orig).



### 3.2.2 Isotopic shifts

The bulk of the obtained δ values scatter around the original isotopic signature ($\delta^{18}O$=-8.53 ‰, $\delta^2H$=-60.7 ‰; see Table 1). The calculated isotopic shifts, $\Delta\delta^{18}O$ and $\Delta\delta^2H$, mostly range between -0.07 and 0.30 ‰ and between -0.2 and 0.7 ‰, respectively (Bottles 1 through 5).

These shifts are rather small. Keeping in mind that the reported external 1σ precisions, ±0.15 ‰ ($\delta^{18}O$) and ±0.6 ‰ ($\delta^2H$), apply to the original water and the water after oven exposure, the encountered shifts practically all lie within the analytical error. This also holds true for the shifts of the closed Bottle 6.

The open Bottle 7, by contrast, showed substantial shifts. After 26 weeks, the $\delta^{18}O$ and $\delta^2H$ values had increased by about 9.23 and 25.5 ‰, respectively.

**4 Discussion**

Following a "Keep It Simple" approach, we have designed an elementary and robust sampler, deliberately avoiding complexity such as pumps, solenoid valves, or remote controls (cf. Coplen et al., 2015; Hartmann et al., 2018; Nelke et al., 2018). Avoiding such complexities greatly reduces the failure risk in the field. We also avoided components that permanently consume power (e.g., normally closed solenoid valves during the "open" phase) and complex tubing systems

that would possibly require parts of the sampling water to be used for tubing flushes. Hence, our collector is relatively easy to reproduce, although some technical skills are required.

Compared to other devices (Coplen et al., 2008; Hartmann et al., 2018), it has fewer but bigger bottles. The latter aspect implies that analyses do not have to be restricted to $\delta^{18}O$ and $\delta^2H$, but other parameters such as $^3H$, major ions (incl. bicarbonate by titration), etc. could in principle be studied as well.

Most importantly, our device gathers integral samples over freely selectable collection periods (minutes to weeks) and effectively reduces post-sampling evaporation. In combination with the low power consumption, these features render it a potential candidate for autonomous rain sampling at remote, unmanned sites. Traditionally, precipitation samples for isotope analyses are mostly collected on a monthly integral basis (e.g., in the Global Network of Isotopes in Precipitation, GNIP; IAEA/WMO, 2019). Yet, due the advent of laser-based isotope analyzers and the associated decreasing costs for analyses

(Berman et al., 2009; Herbstritt et al., 2012; Wassenaar et al., 2018), shorter collection periods (e.g., weekly integral samples) have become more popular (e.g., Otte et al., 2017). An important advantage of sampling schemes with a higher temporal resolution is the gain in flexibility. They allow, for example, a correlation between isotope signature and meteorological variables (Akers et al., 2017; Hughes and Crawford 2013; Rao et al., 2008), but data can still be aggregated to precipitation-weighted monthly values for seamless comparison with other monthly data sets (e.g., GNIP).

Due to the relatively low costs (parts <600 €), several automatic collectors can be deployed simultaneously to address spatial variability. A number of researchers reported pronounced spatial variability of the isotopic composition of rain in small catchments (Fischer et al., 2017; Kato et al., 2013) or at the city scale (Chen et al., 2017). In these studies, $\delta^{18}O$ values partly varied by several ‰ within short distances (a few hundred meters to a few kilometers). In case of throughfall, $\delta^{18}O$ values can even differ significantly on the plot scale, i.e., within meters, and differences of more than 1 ‰ (Allen et al., 2014, 2015;

Kato et al., 2013) or even several ‰ (Hsueh et al., 2016) have been observed. Further, it has been shown that ignoring the spatial variability impairs isotope-based hydrograph separations (Cayuela et al., 2019; Fischer et al., 2017). These findings highlight the need for collector arrays or networks (cf. Lutz et al., 2018; Scholl et al., 1995, 1996). Using expensive, commercial equipment in such networks might be prohibitively expensive for many researchers. Rodgers et al. (2005), for instance, report resource constraints preventing the installation of a second sampler in their catchment to capture the local

altitude effect. In such cases, an affordable custom-made alternative is extremely useful. This also applies to study areas in which vandalism or theft of monitoring equipment might be an issue (cf. Kongo et al., 2010; Otte et al., 2017; Pramana and



Ertsen, 2016). In such settings, researchers on a limited budget would possibly hesitate to leave expensive equipment unattended at a remote site for longer time periods and a low-cost alternative would be appreciated.

Although this work focused on the isotopes $^{18}$O and $^{2}$H, gathered samples could also be analyzed for other isotopes or their hydrochemistry. Since the funnel is permanently exposed, the sampler would act as a bulk sampler (collecting wet and dry deposition). Hydrochemical analyses could, for instance, include chloride for recharge estimations via the Chloride Mass Balance method (e.g., Eriksson and Khunakasem, 1969; Guan et al., 2010).

## 5 Potential modifications

While the presented sampler suits our purposes, we acknowledge that other researchers might have different expectations towards such a collector. Thus, we explicitly encourage others to modify our design and tailor it to their specific needs.

Potential modifications might address the number of bottles and their size. In addition, the tubing can be changed, but we suggest the use of opaque tubing for the exposed section from the funnel to the sampler itself to reduce the risk of algae growth, which has occasionally caused problems in rain collectors (Scholl et al., 1995). Although it might be tempting to replace the push-in ports by ordinary barbed hose fittings, we suggest not to do so for the following reasons: 1) Given the potential risk of disconnection of tubing, the used ports are deemed a safe option. 2) Their handling is easier, particularly when dealing with tens of connections in a confined space. 3) They do not introduce an additional constriction.

Moreover, one could transform the timer-actuated into a volume-controlled rain sampler, for instance by means of a microcontroller-based tipping bucket system. The latter would calculate the filling status of a sampling bottle based on the recorded number of tips and automatically direct the water into the next bottle when the first one is full (cf. Muller et al., 2015).

Also, a transformation into a surface water sampler is feasible. To this end, one could combine the current device with a timer-triggered peristaltic or submersible pump. The microcontroller provides several unused input/output and communication pins and further unutilized resources to allow for such customizations.

## 6 Summary and conclusions

Our microcontroller-based automatic rain sampler enables timer-actuated integral rain sampling. The simple, low-cost device is robust and effectively minimizes post-sampling evaporation from the collection bottles and the associated isotope fractionation. The excellent performance of the device during an extensive evaporation experiment in a laboratory oven (26 weeks; 21-31°C) suggests that even multi-week field deployments in warm climates are feasible. In the spirit of Open Science, we share all relevant details on our sampler and encourage others to adapt it to their specific needs.

## Data availability

All details needed to copy our sampler are freely available (https://www.ufz.de/index.php?en=44048, section Documentation). Data on the evaporation experiment, enabling a performance evaluation, are given in the main article and the corresponding Supplement.

## Author contribution

NM, GL, and TM conceptualized the sampler. GL was responsible for the detailed sampler design and programmed the microcontroller. GL, JF, ABABS, and TM conducted initial tests. NM carried out the evaporation experiment. SMW performed isotope and data analyses. NM wrote the manuscript with contributions from all co-authors.



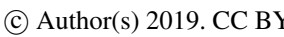

**Competing interests**

The authors declare that they have no conflict of interest.

**Acknowledgements**

This work has been supported by the project "Submarine Groundwater Discharge: Adaption of an Autonomous Aquatic Vehicle for Robotic Measurements, Sampling and Monitoring", funded by The Research Council of Oman (TRC Research Contract No. TRC/RCP/15/001). We would also like to thank the UFZ workshop and Hendrik Zöphel (UFZ) for assistance in construction and Inga Schreiter, Sahand Farhang Darehshouri, and Claudia Cosma (TU Darmstadt) for help during the

evaporation experiment.

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



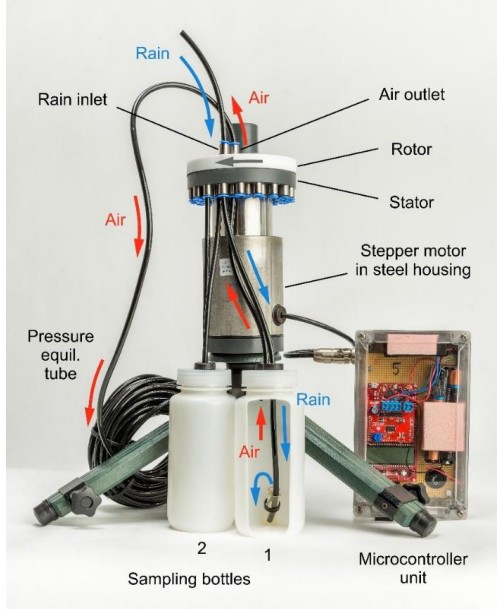

**Figure 1: Overview of the automatic sampler principle (Photo by André Künzelmann, UFZ). Water (blue arrows) flows from the rain inlet through the distribution unit and into the first bottle (cutaway view). Displaced air (red arrows) leaves the bottle and flows through the distribution unit and the air outlet into a 15 m long pressure equilibration tube. At the end of the sampling**
5 **interval, the rotor is turned (grey arrow), leading the rain into the second bottle.**



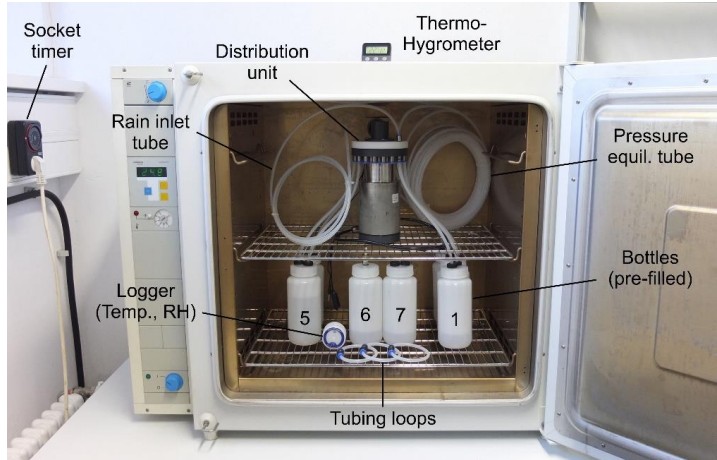

Figure 2: Photograph of the evaporation experiment setup. Note that the bottles were pre-filled with water of known isotopic composition, i.e., no water flowed through the tubing, and the distribution unit did not rotate during the test.





Table 1: Overview of collected samples and associated water losses and isotopic shifts. Water losses Δm are given as absolute mass losses [g] and as percentages of the original water amount [%-orig]. Isotopic signatures ($\delta^{18}O$ and $\delta^{2}H$) and isotopic shifts away from the original composition ($\Delta\delta^{18}O$ and $\Delta\delta^{2}H$) are given in ‰ V-SMOW. Note that the analytical precisions for $\delta^{18}O$ and $\delta^{2}H$
5   account for ±0.15 ‰ and ±0.6 ‰ (±1σ), respectively.

| ID | Description | t [weeks] | Δm [g] | Δm [%-orig] | $\delta^{18}O$ [‰] | $\delta^{2}H$ [‰] | $\Delta\delta^{18}O$ [‰] | $\Delta\delta^{2}H$ [‰] |
|---|---|---|---|---|---|---|---|---|
| Orig. | Original water | 0 | n.a. | n.a. | -8.53 | -60.7 | n.a. | n.a. |
| Bottle 1 | exposed, 100 mL | 6 | 0.17 | 0.17 | -8.32 | -60.2 | 0.21* | 0.5* |
| | | 16 | 0.56 | 0.56 | -8.50 | -60.6 | 0.03* | 0.1* |
| | | 26 | 0.94 | 0.94 | -8.44 | -60.2 | 0.09* | 0.5* |
| Bottle 2 | blocked, 100 mL | 6 | 0.20 | 0.20 | -8.42 | -60.4 | 0.11* | 0.3* |
| | | 16 | 0.56 | 0.56 | -8.23 | -60.0 | 0.30* | 0.7* |
| | | 26 | 0.93 | 0.93 | -8.46 | -60.2 | 0.07* | 0.5* |
| Bottle 3 | blocked, 200 mL | 6 | 0.16 | 0.08 | -8.49 | -60.7 | 0.04* | 0.0* |
| | | 16 | 0.46 | 0.23 | -8.49 | -60.6 | 0.04* | 0.1* |
| | | 26 | 0.81 | 0.40 | -8.59 | -60.9 | -0.06* | -0.2* |
| Bottle 4 | blocked, 300 mL | 6 | 0.43 | 0.14 | -8.53 | -60.8 | 0.00* | -0.1* |
| | | 16 | 1.16 | 0.39 | -8.50 | -60.5 | 0.03* | 0.2* |
| | | 26 | 2.04 | 0.68 | -8.57 | -60.7 | -0.04* | 0.0* |
| Bottle 5 | blocked, 400 mL | 6 | 0.53 | 0.13 | -8.53 | -60.8 | 0.00* | -0.1* |
| | | 16 | 1.47 | 0.37 | -8.55 | -60.7 | -0.02* | 0.0* |
| | | 26 | 2.41 | 0.60 | -8.60 | -60.6 | -0.07* | 0.1* |
| Bottle 6 | closed, 100 mL | 6 | 0.05 | 0.05 | -8.60 | -60.9 | -0.07* | -0.2* |
| | | 16 | 0.14 | 0.14 | -8.65 | -60.9 | -0.12* | -0.2* |
| | | 26 | 0.24 | 0.24 | -8.66 | -60.9 | -0.13* | -0.2* |
| Bottle 7 | open, 100 mL | 6 | 5.27 | 5.27 | -6.85 | -56.1 | 1.68 | 4.6 |
| | | 16 | 14.18 | 14.18 | -3.70 | -46.9 | 4.83 | 13.8 |
| | | 26 | 24.42 | 24.42 | 0.70 | -35.2 | 9.23 | 25.5 |

n.a.: not applicable; *: Δδ within analytical precision ( |±1σ| = 2σ)