# Peer review of "Technical note: A microcontroller-based automatic rain sampler for stable isotope studies"

_Hydrology and Earth System Sciences, 2019_

## Referee Comment (RC1) · Rolf Hut (Referee) · 17 Mar 2019

**Review of "Technical note: A microcontroller-based automatic rain sampler for stable isotope studies" by Michelsen et. al.**

Review by Rolf Hut

I've read the paper "Technical note: A microcontroller-based automatic rain sampler for stable isotope studies" with great interest. I think it is highly relevant for the readership of HESS and recommend publishing it. I only have a minor suggestion that could (hopefully) make the article even more useful to the hydrological community.

**On reproducibility**

I applaud that the authors make their design completely open so the community can reproduce it for their own work. However, I have two issues with the way the material is presented. Firstly, providing a Bill of Materials, technical drawings, software and even an operation manual might not be enough for hydrologist to be able to re-produce the work of the authors. I would strongly recommend to add a "build guide" document to the (impressive) list of documents that the authors already provide. This build guide would detail, step by step, how to construct the sampler. For example, currently it is unclear what fabrication techniques to use in construction of the sampler. I recommend using a site like instructables.com for their comprehensive format and exporting the resulting instructable as a pdf.

**On archiving**

Given the target of the authors "to enable reproduction" and the general nature of institute website designs (and corresponding URLs) to change every season, I would not advice to use an institute website as the main repository for the results of this work. I strongly urge the authors to make all material currently available on https://www.ufz.de/index.php?en=44048 (and the requested build guide, see above) available

- as supplementary material to this publication
- or publish it on Github and create a DOI through Zenodo

**on code, power consumption and re-usability**

I understand the choice for C as a language to program the microcontroller, given the need for low power consumption. I've briefly scanned the code and it is well documented. I particularly like that the section on event based sampling instead of time based sampling is already present, but commented out for now. The C language however has a steep learning curve. Many hydrologist that work in making sensor systems use the Arduino ecosystem for that reason. Since controlling a stepper-motor is the main action on the sampler, I believe

that this could be achieved in the Arduino eco-system, at the cost of greater power consumption. Since there is ample room for a solar panel on top of the sampler, low power might not always be the main concern. I would like to ask the authors to add a sentence or two that the functionality of the sampler could be achieved with parts from the Arduino eco-system as well, although at the cost of greater power consumption.

Concluding, I really like the work presented, recommend it for publication in HESS with only a few minor changes suggested.

**On co-publication**

Finally, a general remark aimed at the editors of HESS, or the leadership of the EGU publishing committee. As I mentioned in my review of Hartmann 2018 ([https://www.hydrol-earth-syst-sci.net/22/4281/2018/hess-22-4281-2018-discussion.html)](https://www.hydrol-earth-syst-sci.net/22/4281/2018/hess-22-4281-2018-discussion.html) I would argue that this work is of interest to both the readership of HESS, as well as of Geoscientific Instrumentation (GI). Shouldn't it be possible to have an article be accepted in both journals? Why not have a single peer review system, followed by a list of papers authors can select from where to publish?

---

## Referee Comment (RC2) · Manfred Groening (Referee) · 1 Apr 2019

Dear colleagues, I have read with interest this interesting manuscript on the design of an automated rain sampler minimising evaporation and therefore ensuring scientifically sound data for stable isotope process studies. The manuscript is well structured and provides a wealth of references to see the state of the art in recent efforts to ensure proper precipitation sampling for stable isotope analysis. I appreciate very much the details revealed by the authors to enable reproduction of such analyser in self-made mode, therefore probably minimising costs for most users having access to a workshop.

The scientific findings are relevant, presented appropriately and comprehensively.

The indicated very low energy consumption and use of cheap commercial batteries is

a key advantage for successful application in many remote sampling scenarios. The setup may not be fully suitable for very cold conditions (anyway not below water freezing point).

The experimental setting is solid and provides evidence of negligible isotopic fractionation due to minimal unavoidable evaporation.

There are only few minor comments to potential users to improve the impact of the paper and minimise problems.

I did not find details on how to connect tubes through the caps into the individual bottles (two connections necessary per cap). This could be seen as very minor issue, but the connection through the cap needs to be completely air tight to atmosphere. One photo would suffice to clarify it.

One minor comment is related to the isolation of bottles after moving the upper disk to the next bottle position as discussed in section 2. This is the moment when each isolated bottle is keeping its actual air pressure at the time of closure, with its internal pressure not anymore being equilibrated via the tubing (page 3, line 15). The experimental data in Table 1 show in general an increased evaporation for hermetically closed bottles versus one bottle open to the atmosphere via a long tube (section 3.2.1, page 4 line 34). This could be caused by the atmospheric pressure fluctuations, resulting in periods of higher or lower pressure in the bottle versus the open atmosphere, and may induce pressure induced air flow and leakages (it is nearly impossible to keep a large area flat sealing pressure tight). The increase of losses with increased water amount could point to a solubility issue (slow penetration of liquid water according to filling height through plastic material).

A second comment is related to the bottle types. I did not find an address for the provider of suitable bottles. However the quality of bottles is of major influence for such study. At the IAEA we have previously (2004) performed long term experiments with nearly 60 different bottle types used for regular water sampling, all filled with same water in triplicate and kept for 6, 12 and 18 months before analysis, recording the weight loss and isotopic shift. After 12 months more than half of bottle types showed evaporation losses above one percent of water weight, associated to delta18O changes of above 0.5 permille. Therefore the proper selection of bottle type is crucial for storage. Glass bottles could be perfect (however even some glass bottle types (!) caused evaporation by imperfect fitting of glass surface to plastic caps), but in most cases high quality HDPE bottles showed best performance at moderate price and robustness.

Overall the paper is of excellent quality and should definitively be accepted. Best regards, Manfred Gröning m.groening@iaea.org

---

## Author Comment (AC1) · 20 May 2019

Dear Dr. Groening,

Thank you very much for your valuable comments on our manuscript. Please find below your reproduced comments, followed by our responses.

COMMENT: Dear colleagues, I have read with interest this interesting manuscript on the design of an automated rain sampler minimising evaporation and therefore ensuring scientifically sound data for stable isotope process studies. The manuscript is well structured and provides a wealth of references to see the state of the art in recent efforts to ensure proper precipitation sampling for stable isotope analysis. I appreciate very much the details revealed by the authors to enable reproduction of such analyser

[Figure]

in self-made mode, therefore probably minimising costs for most users having access to a workshop. The scientific findings are relevant, presented appropriately and comprehensively. The indicated very low energy consumption and use of cheap commercial batteries is a key advantage for successful application in many remote sampling scenarios. The setup may not be fully suitable for very cold conditions (anyway not below water freezing point). The experimental setting is solid and provides evidence of negligible isotopic fractionation due to minimal unavoidable evaporation. There are only few minor comments to potential users to improve the impact of the paper and minimise problems. I did not find details on how to connect tubes through the caps into the individual bottles (two connections necessary per cap). This could be seen as very minor issue, but the connection through the cap needs to be completely air tight to atmosphere. One photo would suffice to clarify it.

RESPONSE: Thank you for this positive and motivating evaluation. Concerning the tubing connections through the bottle caps, we have to admit that this crucial point was indeed not emphasized in our initial manuscript. We use cable grommets that also appeared in the bill of materials, but probably this aspect deserves more attention in the main text. We now added the following sentence: "All tubes are guided through the bottle caps by means of cable grommets (Fig. S2), ensuring a tight connection." In the Supplement, we now provide additional details: "The water and air tubes are guided through the bottle caps by means of cable grommets. The used KAB SNAP 9 cable grommets (see bill of materials) are suitable for tubing diameters between 5.0 and 7.0 mm. They were installed by drilling two 16 mm bores into the bottle caps and pushing the conical part of the grommets through the bores." Moreover, we now include a photograph (see Fig. 1).

COMMENT: One minor comment is related to the isolation of bottles after moving the upper disk to the next bottle position as discussed in section 2. This is the moment when each isolated bottle is keeping its actual air pressure at the time of closure, with its internal pressure not anymore being equilibrated via the tubing (page 3, line

15). The experimental data in Table 1 show in general an increased evaporation for hermetically closed bottles versus one bottle open to the atmosphere via a long tube (section 3.2.1, page 4 line 34). This could be caused by the atmospheric pressure fluctuations, resulting in periods of higher or lower pressure in the bottle versus the open atmosphere, and may induce pressure induced air flow and leakages (it is nearly impossible to keep a large area flat sealing pressure tight). The increase of losses with increased water amount could point to a solubility issue (slow penetration of liquid water according to filling height through plastic material).

RESPONSE: We agree – it is very likely that pressure fluctuations (mostly triggered by the diurnal temperature regime) play a role, particularly if small leakages occur (very likely). Also the filling status of the blocked bottles seems to be relevant in terms of water losses. Hence, we have modified the corresponding paragraph in section 3.2.1 accordingly: "These data suggest that the diffusive loss through the tubing material of the connected bottles (two tubes per bottle, hence 0.28 g; see tubing loop data) is similar to the flux through the bottle material of Bottle 6 (0.24 g). As all connected bottles exhibited greater absolute mass losses, additional leakages, e.g. at the cable grommets in the bottle caps or at the distribution unit, seem likely. In this context, pressure fluctuations, induced by the diurnal temperature regime, probably play a role. It is also noteworthy that the blocked bottles 4 and 5, containing 300 and 400 mL of water, showed the greatest losses (> 2 g). This observation could point towards an influence of the bottle surface area in contact with liquid water on the diffusive water flux through the plastic. Nevertheless, the overall absolute losses are still rather small, particularly when compared to the worst case scenario, an unprotected bottle (Bottle 7)."

COMMENT: A second comment is related to the bottle types. I did not find an address for the provider of suitable bottles. However the quality of bottles is of major influence for such study. At the IAEA we have previously (2004) performed long term experiments with nearly 60 different bottle types used for regular water sampling, all filled

with same water in triplicate and kept for 6, 12 and 18 months before analysis, recording the weight loss and isotopic shift. After 12 months more than half of bottle types showed evaporation losses above one percent of water weight, associated to delta18O changes of above 0.5 permille. Therefore the proper selection of bottle type is crucial for storage. Glass bottles could be perfect (however even some glass bottle types (!) caused evaporation by imperfect fitting of glass surface to plastic caps), but in most cases high quality HDPE bottles showed best performance at moderate price and robustness. Overall the paper is of excellent quality and should definitively be accepted. Best regards, Manfred Gröning m.groening@iaea.org

RESPONSE: In our initially submitted manuscript, we mentioned our supplier, but admittedly quite late – in section 3.1 where we describe the methodology of our evaporation experiment. In section 2 outlining the design, we had tried to keep out suppliers to not impede reading flow and because the section was meant to only describe the principle of the collector. However, we agree that suitable bottles are crucial and hence added the following sentence upon first mention of the bottles, i.e., in the Design section: "With respect to the bottles, we recommend thick-walled HDPE bottles that effectively reduce diffusive water losses (Spangenberg, 2012; personal communication Manfred Gröning, IAEA, https://doi.org/10.5194/hess-2019-93-RC2). For our purposes, we selected 500 mL HDPE wide-mouth bottles by Labsolute (Renningen, Germany; wall thickness approx. 1.7 mm)." Initially we had only cited Spangenberg (2012) in this regard. Since the IAEA has apparently carried out such experiments much earlier, we now also refer to your comment as personal communication and provide the DOI of your review. We hope you agree with this.

Best regards, Nils Michelsen (on behalf of the author team)

[Figure]

**Fig. 1.** Bottle caps with cable grommmets

---

## Author Comment (AC2) · 20 May 2019

Dear Dr. Hut,

Thank you very much for your thoughtful comments on our manuscript. Please find below your reproduced comments, followed by our responses.

COMMENT: I've read the paper "Technical note: A microcontroller-based automatic rain sampler for stable isotope studies" with great interest. I think it is highly relevant for the readership of HESS and recommend publishing it. I only have a minor suggestion that could (hopefully) make the article even more useful to the hydrological community.

RESPONSE: Thank you for this positive and motivating evaluation. We are confident that your suggestions indeed improved our manuscript.

[Figure]

COMMENT: On reproducibility: I applaud that the authors make their design completely open so the community can reproduce it for their own work. However, I have two issues with the way the material is presented. Firstly, providing a Bill of Materials, technical drawings, software and even an operation manual might not be enough for hydrologist to be able to re-produce the work of the authors. I would strongly recommend to add a "build guide" document to the (impressive) list of documents that the authors already provide. This build guide would detail, step by step, how to construct the sampler. For example, currently it is unclear what fabrication techniques to use in construction of the sampler. I recommend using a site like instructables.com for their comprehensive format and exporting the resulting instructable as a pdf.

RESPONSE: We agree that the assembly of the collector parts is not fully self-explanatory and that a build guide is a useful addition to the material included in our initial submission. Hence, we have compiled such a build guide (see attached file) and hope that it will facilitate the construction for others.

COMMENT: On archiving: Given the target of the authors "to enable reproduction" and the general nature of institute website designs (and corresponding URLs) to change every season, I would not advice to use an institute website as the main repository for the results of this work. I strongly urge the authors to make all material currently available on https://www.ufz.de/index.php?en=44048 (and the requested build guide, see above) available - as supplementary material to this publication - or publish it on Github and create a DOI through Zenodo

RESPONSE: We now include all mentioned material in the supplement of the paper (instead of using github) to ensure that interested readers can obtain the paper and all related data from a single source. That said, we will maintain the initially mentioned website (https://www.ufz.de/index.php?en=44048) – not as an exclusive primary data source, but to announce updates, new developments, etc. As we have been contacted and asked for 3D files (for 3D printing of selected parts), we now also provide .stl files in the supplement.

COMMENT: on code, power consumption and re-usability: I understand the choice for C as a language to program the microcontroller, given the need for low power consumption. I've briefly scanned the code and it is well documented. I particularly like that the section on event based sampling instead of time based sampling is already present, but commented out for now. The C language however has a steep learning curve. Many hydrologist that work in making sensor systems use the Arduino ecosystem for that reason. Since controlling a stepper-motor is the main action on the sampler, I believe that this could be achieved in the Arduino eco-system, at the cost of greater power consumption. Since there is ample room for a solar panel on top of the sampler, low power might not always be the main concern. I would like to ask the authors to add a sentence or two that the functionality of the sampler could be achieved with parts from the Arduino eco-system as well, although at the cost of greater power consumption. Concluding, I really like the work presented, recommend it for publication in HESS with only a few minor changes suggested.

RESPONSE: We also have the impression that Arduino microcontrollers are rather popular in the community. Hence, we followed your suggestion and added the following sentence in section 5 (Potential modificstions): "Although we used a Texas Instruments microcontroller, we can imagine that the functionality of the sampler could also be achieved with parts from the popular Arduino ecosystem, though probably at the cost of greater power consumption."

Best regards, Nils Michelsen (on behalf of the author team)

Please also note the supplement to this comment:
https://www.hydrol-earth-syst-sci-discuss.net/hess-2019-93/hess-2019-93-AC2-supplement.pdf

—————————————————

[Figure]

**Supplement:**

**Build guide**

A cross-section of the distribution unit to be assembled in given in Figure 1. The technical drawing provides an overview of the used parts, for which also individual drawings are available.

[Figure]

**Figure 1** Schematic cross-section giving an overview of the main parts of the distribution unit

1. Prepare the rotor (PTFE) and the stator (PVC-U) discs according to the corresponding technical drawings, e.g. with a lathe, and drill the required bores, e.g. with a drilling press. Cut the required threads into the material with a tap.

2. Screw the push-in ports into the two discs (2 into the rotor and 36 into the stator). Do so carefully – the threads in plastic, especially those in the rotor (PTFE), are rather sensitive.

3. Place the adhesive side of the neoprene rubber seal onto the stator. Make sure that the bores in the two parts are aligned.

4. Prepare the two parts of the motor housing (e.g. turn the components from stainless steel on a lathe) according to the corresponding technical drawing and mount them together (e.g. by welding). Drill a 16 mm bore into the lower part of the motor housing.

5. Prepare the motor housing lid (PVC-U) according to the corresponding technical drawing, e.g. on a lathe.

6. Prepare the spring housing (PVC-U), the spring guide (steel), and the spring washer (steel) according to the technical drawings, e.g. on a lathe.

The prepared parts should look similar to those shown below (Figure 2).

[Figure]

**Figure 2** Overview of the main parts of the distribution unit (Photo by André Künzelmann, UFZ)

7. Assemble the parts according to the technical drawing of the entire distribution unit (Figure 1). First mount the stator, the motor housing, and the stepper motor together (four M5x12 screws). Do not forget the O-rings.

8. Add the rotor and fix it with the spring array consisting of the spring washer, the spring, the spring guide, and the corresponding screw (M4x35). With the spring, the sealing pressure can be adjusted. It has an initial length of 40 mm and a rate of 4.94 N/mm.

[Figure]

**Figure 3** Spring for sealing pressure adjustment

Setting the spring to a length of 25 mm (see Figure 3) creates the required sealing force of (40 mm - 25 mm) · 4.94 N/mm = 74.1 N.

The sealing area is roughly 5945 mm². Hence, the sealing pressure is 74.9 N / 5945 mm² = 0.0125 N/mm² or 1.275 m of water column. As a rule of thumb, the pressure that is applied to the sealing is equal to the water pressure that it will withstand without leakage. Note: Actually, there should be no water pressure applied to the sealing at all – as long as the system works as intended, all water flows directly into the sampling bottles. Finally, mount the spring housing onto the rotor with four M5x10 screws.

9. Mount the cable grommet into the 16 mm bore of the motor housing and push the cable through the grommet. Connect the cable strands to the strands of the stepper motor with butt connectors (see Figure 4). Connect the round connector to the other end of the cable.

10. Close the motor housing with the lid. The O-ring of the lid tends to squeeze out of its place when too much force is applied. Take care to keep it in place or squeeze it back.

[Figure]

**Figure 4** Distribution unit, view from below

At this point, the distribution unit should look like shown in Figure 5.

[Figure]

**Figure 5** Assembled distribution unit (Photo by André Künzelmann, UFZ)

11. Prepare the parts of the control unit according to Figure 6. Mount the coupler into the enclosure and the pin headers, the incremental encoder, and the rocker switch onto the prototyping circuit board.

[Figure]

**Figure 6** Overview of the main parts of the control unit (Photo by André Künzelmann, UFZ)

12. Assemble the parts according to the provided circuit diagram and follow the manual for software upload.

The assembled control unit should look like shown in Figure 7. When closing the enclosure, small pieces of foam can be used to keep the battery holders in place (see also Figure 1 in the main text). Moreover, it is often advisable to place a small desiccant pack into the control unit.

[Figure]

**Figure 7** Assembled control unit (Photo by André Künzelmann, UFZ)

13. Connect the distribution unit and the control unit with the cable.

Now, the sampling bottles have to be prepared and connected.

14. Drill two 16 mm bores into each bottle cap and mount the cable grommets.

15. Connect the HDPE bottles to the distribution unit with short pieces of LDPE tubing. Water tubes lead from the push-in ports of the outer circle in the stator to the bottom of the sampling bottles. Thus, they have to be longer than the air tubes. The latter are pushed into the ports of the inner circle and extend just below the bottle caps. To remove a tube from a push-in port, the blue ring of the port has to be pushed.

16. Connect the pressure equilibration tube (LDPE; 15 m) to the inner port in the rotor.

17. Connect a rain inlet tube to the outer port in the rotor. The other end of the tube is connected to the rain funnel. To this end, we usually shortened the outlet of our plastic funnels until we could screw in an additional push-in port (R 1/2" thread). The latter can also be secured with electrical tape.

Now, the device can be operated (see separate manual).